# Biocompatibility, Biomineralization and Induction of Collagen Maturation with the Use of Calcium Hydroxide and Iodoform Intracanal Dressing

**DOI:** 10.3390/jfb14100507

**Published:** 2023-10-10

**Authors:** Carlos Roberto Emerenciano Bueno, Jimena Lama Sarmiento, Ana Maria Veiga Vasques, Ana Cláudia Rodrigues da Silva, Luciano Tavares Angelo Cintra, João Miguel Marques Santos, Eloi Dezan-Júnior

**Affiliations:** 1Endodontic Section, Department of Preventive and Restorative Dentistry, Araçatuba School of Dentistry (FOA), São Paulo State University (Unesp), Araçatuba 16015-050, SP, Brazilanavvasques@hotmail.com (A.M.V.V.);; 2Institute of Endodontics, Faculty of Medicine, University of Coimbra, 3004-531 Coimbra, Portugal

**Keywords:** calcium hydroxide, collagen, endodontics, inflammation, iodoform, intracanal dressing

## Abstract

Biocompatibility and biomineralization of root canal dressings are important requirements for periapical healing. This study evaluated the inflammatory response, biomineralization and tissue repair by collagen fiber maturation in the subcutaneous tissue of rats. Eighteen Wistar rats (*n* = 6) received subcutaneous implants: calcium hydroxide + propylene glycol [CH+P], calcium hydroxide + propylene glycol + iodoform [CH+P+I], iodoform + carbowax [I+Cwax] and carbowax [Cwax]. Extra empty tubes were used as a control [C]. After 7, 15 and 30 days, the implants were removed with surrounding tissue for staining of hematoxylin-eosin, Von Kossa, picrosirius red and without staining for analysis under polarized light. Results were analyzed via Kruskal–Wallis followed by Dunn testing for nonparametric data and ANOVA followed by a Tukey post hoc test for parametric data (*p* < 5%). At 7 days, all groups showed a moderate inflammatory reaction and thick fibrous capsule, except the [Cwax] group, with a severe inflammatory infiltrate (*p* < 0.05). After 15 days, all groups but control had a decrease in inflammatory response. At 30 days, all groups presented a mild reaction and thin fibrous capsule (*p* > 0.05). Only groups containing calcium hydroxide were found to be positive using Von Kossa staining and polarized light in all periods. At 7 days, all groups showed a higher proportion of immature fibers. At 15 days, the [CH+P] and [Cwax] groups increased their proportion of mature/immature fibers. At 30 days, only the [CH+P] group presented a significant prevalence of mature collagen fibers (*p* < 0.05). All groups showed biocompatibility, but only groups containing calcium hydroxide induced biomineralization. The addition of iodoform delayed tissue healing.

## 1. Introduction

The main objective of endodontic therapy is to cure or prevent apical periodontitis, which still shows a high prevalence in the world population [1]. In cases of root canal infections and pulp necrosis, endodontics aims to reduce microorganisms and prevent the re-infection of root canal systems [2]. The microbiota load reduction in infected teeth is achieved by means of biomechanical preparation, which involves an effective root canal cleaning and shaping with endodontic files instrumentation and chemical irrigation. However, due to the complexity of the root canal system, areas such as isthmus or lateral canals are not touched by the mechanical action of the instruments, nor do they come in contact with irrigants, hampering the efficacy of the reduction of bacterial load during root canal preparation [3,4,5]. Therefore, the complementary use of an intracanal dressing is an additional step that may help clinicians in controlling intrarradicular infection and preventing interappointment microbial ingress or recontamination [4,6,7].

Calcium hydroxide (CH) may be considered one of the oldest biomaterials used in endodontics, having been introduced by Hermann in 1920 in Germany [8]. CH, obtained by the hydration of calcium, is widely used in clinical situations involving necrotic pulp due to its antimicrobial and biologic effects, which are linked to the dissociation of calcium hydroxide particles, allowing the diffusion of calcium and hydroxyl ions into the dentinal tubules, isthmus and periapical tissue, improving the antiseptic and repair action [9,10,11]. Clinically, calcium hydroxide is manipulated with a vehicle (viscous, oily or aqueous) and may be added with iodoform as a radiopacifier agent, to facilitate the visualization of an adequate root canal filling of the dressing [12].

The iodoform, besides acting as a radiopacifier, can show antimicrobial activity by release of iodine [13,14]. Formulations using calcium hydroxide associated with iodoform have demonstrated antimicrobial efficacy against microbial strains present in periapical lesions [13,15]. Based on this antimicrobial feature, previous authors proposed the use of iodoform and carbowax (a polyethylene glycol with a higher molecular mass, as inert vehicle), without calcium hydroxide, as an intracanal dressing [16,17]. This association was later commercially available on the market in early 2022, regulated by the Brazilian National Health Surveillance Agency (Register No 80013980041) [18] with the manufacturer indication of filling entire root canal and extrusion to periapical tissue [19]. Although biocompatibility of this association was reported in rats [16], it was performed on superficial wounds, in a model noncompliant with ISO standard tests for biocompatibility evaluation [20]. In addition, endodontists speculate that a lack of a biomaterial such as calcium hydroxide in an intracanal dressing could delay or jeopardize the deposition of mineralized tissue or the posttreatment healing process in periapical tissues.

Thus, the objective of this study was to evaluate the inflammatory response, deposition of mineralized tissue and potential of tissue repair by collagen fiber maturation in subcutaneous tissue of rats induced by iodoform, calcium hydroxide pastes and their associations. The null hypothesis was that there would be no significant difference between iodoform and calcium hydroxide pastes regarding inflammation, biomineralization and collagen fiber maturation.

## 2. Materials and Methods

### 2.1. Animals

The sample size used in this study (six animals per period/*n* = 6) was based on previous studies of biocompatibility of endodontic materials in the subcutaneous tissue of rats, using power sample 90% and considering alpha error of 0.05 to recognize a significant difference; a minimum number of six rats per analysis period was indicated as minimum necessary [21].

This study was approved by the local institutional Ethics Committee on the Use of Animals at Araçatuba School of Dentistry (CEUA protocol 00425-2018) and conducted in accordance with the ARRIVE (Animal Research: Reporting of in Vivo Experiments) guidelines [22]. All surgery was performed under anesthesia and all efforts were made to minimize animal suffering.

Eighteen 3-month-old male Wistar rats, weighing 250–300 g, were used in this study, housed in temperature-controlled rooms (23 ± 2 °C) and kept under 12 h light/dark cycle, receiving solid diet (Guabi Nutrilabor, Mogiana Alimentos SA, Campinas, Brazil) and water ad libitum.

Ninety polyethylene tubes (Abbott Laboratories of Brazil, Sao Paulo, Brazil) with 1.5 mm external diameter, 1.0 mm internal diameter and 10 mm length were used [23,24] and filled with the following test materials: calcium hydroxide powder P.A. (Pro Analysi, Biodinâmica, Ibiporã, PR, Brazil) + propylene glycol (Quimidrol, Joinville, SC, Brazil) (1:1) [CH+P]; calcium hydroxide powder P.A. + propylene glycol + iodoform (Biodinâmica, Ibiporã, PR, Brazil) (2:1:1) [CH+P+I]; iodoform + carbowax (Polystar Comércio de Essências Ltd., Sao Paulo, Brazil) (5:1) [I+Cwax]; and carbowax [Cwax]. Extra empty tubes were used as control group [C].

The intracanal dressing pastes were prepared by means of manual spatulation with sterilized spatula No. 24 and a glass plate (for each group), with the aforementioned proportions, until paste was homogenous. Then, pastes were inserted into the tubes with the aid of a Lentullo drill (FKG Dentaire, La Chaux-de-Fonds, Switzerland) in an electric motor (VDW Gold, VDW Dental, Munich, Germany), under 1200 RPM.

### 2.2. Surgical Procedures

After administration of xylazine (10 mg/kg) and ketamine (25 mg/kg) intramuscular anesthesia, the backs of the animals were shaved, antisepsis was obtained with 5% iodine solution and a 2 cm incision was performed in a head–tail orientation with No. 15 scalpel blade (Bard-Parker, Dickinson & CO., Franklin Lakes, NJ, USA). The skin was reflected to create two pockets on the right side (upper and lower) and two other pockets on the left side of the incision (upper and lower). Four polyethylene tubes containing the described materials, along with an empty tube, were implanted in the dorsal region of each animal [25] and the skin was sutured with a 4-0 nylon suture (Shalon Suturas, Montes Belos, GO, Brazil).

### 2.3. Histological Analysis

After 7, 15 and 30 days, animals were euthanized by an anesthetic overdose. Polyethylene tubes were removed with the surrounding tissues, fixed in 10% formalin solution, processed and embedded in paraffin followed by, serially, 5 μm cuts for staining with hematoxylin-eosin (HE) and picrosirius red (PSR). Biomineralization was assessed with 10 μm cuts stained with Von Kossa (VK) technique, used to observe mineralization, as it darkly stains mineralized structures and other sections remain unstained, for examination under polarized light (PL) to observe the presence of birefringent structures [23,26]. A single calibrated operator performed histologic analysis at the tube opening in a blinded manner under ×400 light microscopy (DM 4000 B; Leica, Wetzlar, Germany). Tissue reactions at the lower center of the polyethylene tube were scored according to previous studies [23,24] as: 0, few inflammatory cells or no reaction; 1, less than 25 cells and mild reaction; 2, between 25 and 125 inflammatory cells and moderate reaction; and 3, 125 or more inflammatory cells and severe reaction. Fibrous capsule was considered thin when <150 μm and thick when >150 μm. Biomineralization was considered as positive or negative, according to VK and PL [23,26].

The maturation level of collagen fibers was analyzed by PSR under PL microscopy (DM 4000 B; Leica, Wetzlar, Germany). The program QWin was used (×400 magnification; Leica QWin V3; Leica Microsystems, Wetzlar, Hessen, Germany), allowing the selection of corresponding colors for each type of collagen fiber. After color selection, the program automatically calculated the marked area of each collagen type. Greenish-yellow fibers are considered immature and thin, whilst yellowish-red fibers are considered mature and thick [23,26].

The values of red and green fiber obtained by the QWin program were converted into percentages in relation to the total area. Then, the percentage of mature fibers was divided by the percentage of immature fibers, in order to obtain the mature/immature fibers proportion. Thus, values close to 1 (or higher) indicate a predominance of mature fibers (more than 50%), whereas values lower than 1 indicate a prevalence of immature fibers.

### 2.4. Statistical Analysis

Data were collected and analyzed by a single, calibrated and blinded operator. The SigmaPlot 12.0 (Systat Software Inc., San Jose, CA, USA) program was used for the statistical analysis. The normal distribution of data was confirmed by the Shapiro–Wilk test. The parametric data were analyzed by one-way analysis of variance (ANOVA), followed by the Tukey post-hoc test. Nonparametric data were analyzed using the Kruskal–Wallis test, followed by the Dunn test. The *p*-value was considered significant at 5%.

## 3. Results

### 3.1. Control Group [C]

A moderate inflammatory reaction was observed in the 7- and 15-day period (Figure 1A,B and Table 1). The inflammatory infiltrate present was composed generally of lymphocytes and macrophages in a thick fibrous capsule. After 30 days, the capsule surrounding the opening of the tube decreased in thickness and exhibited a lower inflammatory infiltrate (Figure 1C). The control group was negative for VK staining and no birefringent structures were observed under PL (Figure 2a–c).

In terms of the amount of collagen fibers, there was a higher proportion of immature than mature fibers on days 7 (0.619), 15 (0.805) and 30 (0.724) (Table 2).

### 3.2. Calcium Hydroxide + Propylene Glycol Group [CH+P]

On days 7 and 15, a moderate inflammatory reaction (Figure 1D,E and Table 1) was present, then it reduced until day 30 (Figure 1F,f and Table 1). The fibrous capsule at the opening of the tube was thick in the first two periods, and became thin by the end (Figure 1d,e and Table 1). VK staining was positive and birefringent structures were observed under LP in all analyzed periods (Figure 2D,d,E,e,F,f).

PSR staining showed a prevalence of immature fibers at 7 and 15 days and a notable increase of mature collagen fibers at day 30 (Figure 3D–F and Table 2), with the highest proportion of mature fibers in comparison with the other experimental groups (*p* < 0.05).

### 3.3. Calcium Hydroxide + Propylene Glycol + Iodoform Group [CH+P+I]

The two first analyzed periods presented a moderate inflammatory reaction with a thick inflammatory fibrous capsule with macrophages, lymphocytes, giant cells and some large blood vessels profiles (Figure 1G,g,H,h). On day 30, the intensity of inflammation was reduced in 50% of samples (Figure 1I and Table 1) and the fibrous capsule was thin, similar to the control group (Figure 1i). VK staining was positive and birefringent granulations were present in all analyzed periods under PL (Figure 2G,g,H,h,I,i).

In all experimental periods, there was a prevalence of immature fibers (Figure 3 G–I; Table 2).

### 3.4. Iodoform + Carbowax Group [I+Cwax]

A moderate inflammatory reaction was observed in the 7- and 15-day periods. The inflammatory infiltrate was composed generally of macrophages and some giant cells contained in a fibrous capsule, which was thick (Figure 1J,j,K,k and Table 1). At the end of the experiment, the fibrous capsule reduced its thickness, presenting a mild inflammatory reaction in three out of six specimens (Figure 2L,l and Table 1). The other samples presented moderate to severe inflammatory reactions (Figure 1L,l and Table 1). VK staining was negative and no birefringent structures were observed under LP in all analyzed periods (Figure 2J,j,K,k,L,l).

There was a prevalence of immature collagen fibers in all periods, mainly on day 15 (Figure 3J–L and Table 2).

### 3.5. Carbowax Group [Cwax]

This experimental group was the one that presented the highest inflammatory reaction on day 7 (*p* < 0.05), consisting of macrophages and lymphocytes in a thick fibrous capsule (Figure 1M,m and Table 1). On day 15, there was a reduction of the inflammatory reaction (score 2) and the thickness of the fibrous capsules in half of the specimens was classified as thin (Table 1 and Figure 1N,n). At the end of the experiment, a mild inflammatory reaction was present and the fibrous capsules of all analyzed specimens were thin (Table 1 and Figure 1O,o). VK staining was negative and no birefringent structures were observed under LP in all analyzed periods (Figure 2 M,m,N,n,O,o).

Immature collagen fibers appeared in a higher proportion during all the analyzed periods, especially at 30 days (Figure 3M–O).

### 3.6. Comparison among Groups

All the groups, excluding the carbowax, exhibited a moderate inflammatory reaction (Table 1) and a thick fibrous capsule on day 7, similar to the control group, which is normal due to the trauma produced by the surgical process (*p* > 0.05). At 15 days, the fibrous capsule of some groups became thin, and on day 30 there was observed a mild reaction and a thin fibrous capsule in all groups, the same as the control, which demonstrated the biocompatibility of all the pastes (*p* > 0.05). Only groups with calcium hydroxide in their composition presented mineralized tissue at the openings of the tubes with the Von Kossa staining and under polarized light in all analyzed periods. The PSR analysis showed that all groups had a higher proportion of immature fibers at 7 days, since all the values for this period were <1 (Table 2). At 15 days, the control, [CH+P] and [Cwax] group had an increase in the mature/immature fiber proportion, unlike the other groups. At 30 days, [CH+P] was the only group that had a prevalence of mature fibers, presenting a significant difference with all the others (*p* < 0.05).

## 4. Discussion

This study evaluated the tissue response to a biomaterial (calcium hydroxide), iodoform and its association regarding biocompatibility, biomineralization and collagen fibers. Based on results, the null hypothesis was partially rejected, since all groups presented biocompatibility but only pastes containing a biomaterial (calcium hydroxide) allowed biomineralization and superior collagen fiber maturation.

The tissue repair process after root canal treatment consists of a series of biological events that begin with the inflammation and end with a tissue proliferation and remodeling phase [27], which was observed in our study. All groups exhibited a moderate inflammatory reaction on day 7, including the control group, which is attributed to the trauma produced by the surgical process [23]. Cintra et al. [24] evaluated the tissue response of different endodontic materials and observed that the main histological criterion regarding biocompatibility is the reduction of inflammation over time, corroborating the present results, which include a mild tissue reaction and a thin fibrous capsule in all the groups at the end of the experiment.

Although root canal dressings should have antibacterial properties, they should also stimulate tissue repair by the deposition of mineralized tissue [13,28]. Early studies of Holland et al. [29] histologically evaluated the healing process of teeth with open apices, concluding that the presence of calcium hydroxide in the intracanal dressing favored the deposition of mineralized tissue in the periapical region. The importance of a biomaterial addition to endodontic materials was also pointed by Bueno et al. [23], who investigated resinous endodontic sealers containing calcium hydroxide, and observed that sealers with more CH liberation showed enhanced biomineralization. In this study, the addition of iodoform to the calcium hydroxide paste did not interfere in the mineralization process, in accordance with previous studies [14,29,30], which evaluated the effect of radiopacifying agents on the pH level and calcium ion release.

On the other hand, the iodoform and carbowax paste did not induce deposition of mineralized tissue in any analyzed periods. Previous reports suggested the need for mineralized tissue deposition after endodontic treatment in order to promote a biological sealing [31,32].

In this study, the presence of collagen fibers is a positive signal that indicates the repair of a traumatized area. The analysis of collagen fibers found in each period was performed by observing picrosirius red (PSR) staining. This is a specific method for collagen fiber detection, which is capable of distinguish different collagen fiber types, especially when this protein is present in small amounts or is too thin. PSR is an elongated dye molecule that reacts with collagen, increasing normal birefringence [33], allowing distinction of the differences in fiber color: greenish-yellow colors suggest that the collagen is poorly packed (immature fibers), whilst yellowish-red colors are originated from tightly packed fibers (mature fibers) [23].

At 7 days, all the groups presented a prevalence of immature collagen fiber, the same as the control group. After 15 days, it was expected to have a greater proportion of mature/immature fiber. However, the [I+C] group showed an evident decrease of this proportion. This may be explained by the tissue being in contact with an irritating material; poorly packed fibers (immature fibers) are the first to be affected and continue to be renewed; the tightly packed fibers (mature fibers) are more resistant to the aggression and appear in greater proportion when fiber renewal subsides due to a decrease in the inflammatory stimulus. When the production of collagen fibers restarts, there is a greater evidence of immature young collagen fibers, which was observed for the [I+C] and [Cwax] groups.

At 30 days, [CH+P] was the only group that had a prevalence of mature fibers, presenting a significant difference with the others (*p* < 0.05) (Table 2). The addition of iodoform to the calcium hydroxide paste [CH+P+I] group hindered tissue repair and delayed the maturation of collagen fibers. Estrella et al. [13] considered that the biological activity of iodoform may only be considered a hypothesis, due to lack of investigations to demonstrate it. They also speculated that an addition of iodoform to the calcium hydroxide paste would only enhance the radiopacity, which is in agreement with our results. Moreover, a study performed with implants in subcutaneous tissue of rats conducted by de Morais et al. [34] evaluated the tissue response by hematoxylin-eosin staining to Portland cement added with iodoform, concluding that the paste was harmless to connective tissue.

The different mechanisms and properties of two different associated medicaments could minimize the action of one of them, instead of potentializing their effects. In this sense, it could be inferred that the only benefit of the iodoform addition would be to enhance the radiopacity of calcium hydroxide when used as intracanal dressing, in agreement with the studies of Lourenço-Neto et al. [35] and Marques et al. [36], which evaluated clinical and radiographic outcomes of Portland cement added to radiopacifying agents.

The action mechanism of calcium hydroxide [Ca(OH)_2_] paste is related to the ionic dissociation of calcium (Ca^2+^) and hydroxyl (OH^-^) ions, resulting in a higher pH (an antimicrobial property) and activation of alkaline phosphatase, triggering the biomineralization phenomena [9,23]. The Ca ions linking to carbon dioxide (CO_2_) from the tissues results in the formation of calcite crystals (calcium carbonate—CaCO_3_), birefringent to PL, and induces the formation of calcified areas by serving as a mineralization nucleus [23,26]. Consequently, the link of calcium and formation of CaCO_3_ reduces the presence of local carbon dioxide, which is used by bacteria for anaerobic respiration [9,37], providing an additional explanation for the antimicrobial properties exhibited by this intracanal medication.

Clinically, besides conventional root canal therapy with necrotic pulp, intracanal dressing with calcium hydroxide has also been recommended by the American Association of Endodontists (AAE) and the European Society of Endodontology (ESE) in cases of regenerative endodontic procedures aiming for root decontamination by the first appointment for 2–4 weeks [38,39,40], as a treatment of immature teeth with nonvital pulp in adults [41,42]. A retrospective study conducted by Bose et al. [43] evaluated the radiographic outcomes in immature teeth with necrotic pulp treated with regenerative endodontic procedures. Authors highlighted that the use of calcium hydroxide and triple antibiotic paste as intracanal medicament improves further development of the pulp–dentin complex, and better outcomes were observed when medication remained within the root canal.

In biocompatibility analysis, the fibrous capsule was also evaluated. In all specimens, fibrous capsules surrounding the tubes were initially thick and became thin by the end of the experimental period, evidence of biocompatibility and major inflammation resolution [23,26,44]. Present findings of biocompatibility corroborate a previous study, in which the addition of calcium hydroxide to endodontic sealer showed important effects on tissue healing, accelerating the process of repair [45]. Carbowax presented the highest levels of immature fibers at 30 days, which does not indicate lack of biocompatibility, but could indicate a delayed tissue healing reaction. This result does not fully support the manufacturer indication of extrusion to periapical tissue and requires more studies.

The animal model used shows an advantage, because subcutaneous connective tissue does not produce biological hard tissues (such as dentin, cementum and bone) that could hinder calcium salt detection, evidencing the tissue response to test material. These results do not reproduce a complete analysis of the reactions that may occur in human conditions, but they are significant for a preliminary assessment of biocompatibility and biomineralization. However, as in any preclinical experimental research, present findings should be interpreted with caution before being extrapolated to clinical conditions.

Therefore, it is important to highlight the limitations of this research; there is a difference between the experimental (animal model) and a clinical environment, where the presence of dentin could act as buffering agent, neutralizing the alkaline environment. Also, investigation of physico-chemical properties (solubility, radiopacity) and additional accurate assessment of inflammation and biomineralization with immunomarkers should be performed to complement these findings.

## 5. Conclusions

According to the results observed in this animal model, all groups showed adequate biocompatibility. Only pastes containing a biomaterial of Ca(OH)_2_ were able to induce biomineralization. The addition of the iodoform to the calcium hydroxide paste delayed collagen fibers’ maturation.

## Figures and Tables

**Figure 1 jfb-14-00507-f001:**
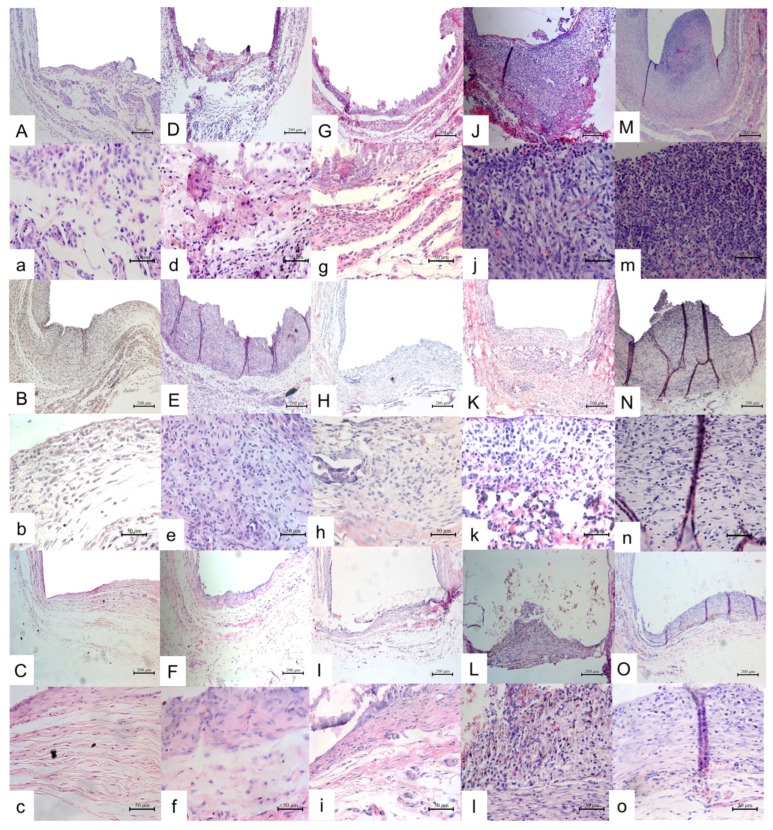
Tissue reaction on experimental groups: [C] group: (**A**–**C**) (days 7, 15 and 30 HE, ×100) and (**a**–**c**) (days 7, 15 and 30, HE, ×400); [CH+P] group: (**D**–**F**) (days 7, 15 and 30, HE, ×100) and (**d**–**f**) (days 7, 15 and 30, HE, ×400); [CH+P+I] group: (**G**–**I**) (days 7, 15 and 30, HE, ×100) and (**g**–**i**) (days 7, 15 and 30, HE, ×400); [I+Cwax] group: (**J**–**L**) (days 7, 15 and 30, HE, ×100) and (**j**–**l**) (days 7, 15 and 30, HE, ×400); [Cwax] group: (**M**–**O**) (days 7, 15 and 30, HE, ×100) and (**m**–**o**) (days 7, 15 and 30, HE, ×400).

**Figure 2 jfb-14-00507-f002:**
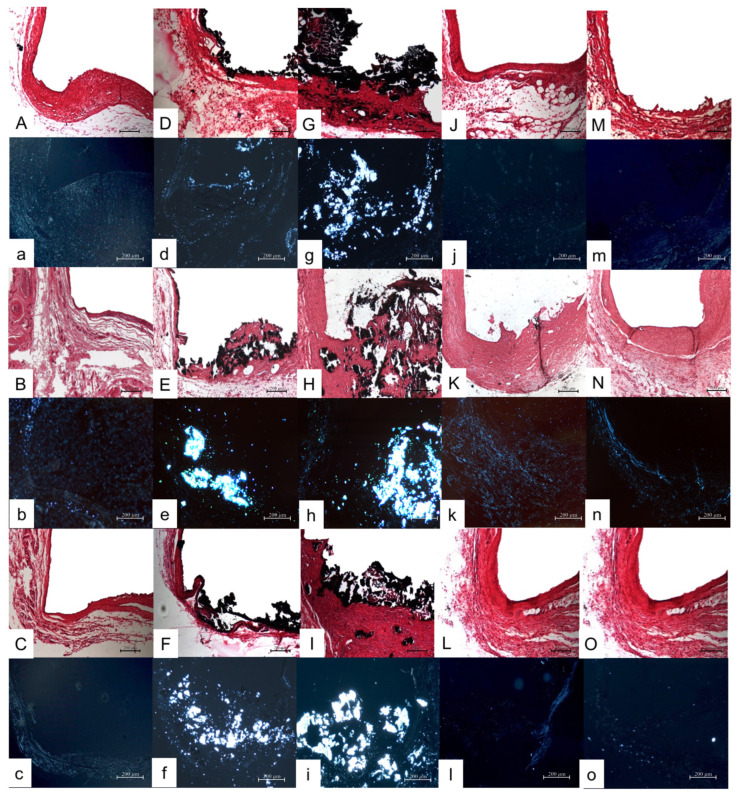
Mineralization on experimental groups: [C] group: (**A**–**C**) (days 7, P and 30; Von Kossa, ×100) and (**a**–**c**) (days 7, 15 and 30, polarized light, ×100); [CH+P] group: (**D**–**F**) (days 7, 15 and 30; Von Kossa, ×100) and (**d**–**f**) (days 7, 15 and 30; polarized light, ×100); [CH+P+I] group: (**G**–**I**) (days 7, 15 and 30; Von Kossa, ×100) and (**g**–**i**) (days 7, 15 and 30; polarized light, ×100); [I+Cwax] group: (**J**–**L**) (days 7, 15 and 30; Von Kossa, ×100) and (**j**–**l**) (days 7, 15 and 30, polarized light, ×100); [Cwax] group: (**M**–**O**) (days 7, 15 and 30; Von Kossa, ×100) and (**m**–**o**) (days 7, 15 and 30, polarized light, ×100).

**Figure 3 jfb-14-00507-f003:**
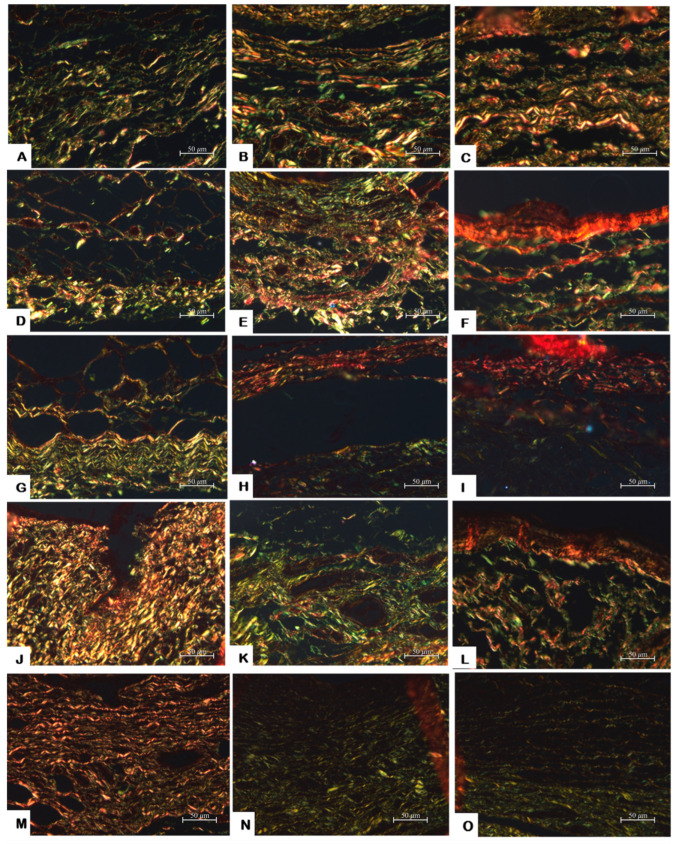
Collagen fibers maturation: [C] group: (**A**–**C**) (days 7, 15 and 30 HE, ×400); [CH+P] group: (**D**–**F**) (days 7, 15 and 30, HE, ×400); [CH+P+I] group: (**G**–**I**) (days 7, 15 and 30, HE, ×400); [I+Cwax] group: (**J**–**L**) (days 7, 15 and 30, HE, ×400); [Cwax] group: (**M**–**O**) (days 7, 15 and 30, HE, ×400).

**Table 1 jfb-14-00507-t001:** Number of samples from each group with inflammatory scores and median; fibrous capsule thickness and mineralization. Different superscript letters indicate statistical difference.

	Inflammatory Score				Median	Fibrous Capsule		Biomineralization	
	0	1	2	3		Thick	Thin	VK	PL
7 days									
C	0	1	4	1	2 ^a^	6	0	**−**	**−**
CH+P	0	1	4	1	2 ^a^	6	0	**+**	**+**
CH+P+I	0	0	5	1	2 ^a^	6	0	**+**	**+**
I+Cwax	0	0	4	2	2 ^a^	6	0	**−**	**−**
Cwax	0	0	2	4	3 ^b^	6	0	**−**	**−**
15 days									
C	0	2	3	1	2 ^a^	6	0	**−**	**−**
CH+P	0	1	5	0	2 ^a^	4	2	**+**	**+**
CH+P+I	0	1	4	1	2 ^a^	3	3	**+**	**+**
I+Cwax	0	0	4	2	2 ^a^	5	1	**−**	**−**
Cwax	0	0	4	2	2 ^a^	3	3	**−**	**−**
30 days									
C	1	5	0	0	1 ^a^	0	6	**−**	**−**
CH+P	0	4	2	0	1 ^a^	0	6	**+**	**+**
CH+P+I	0	3	3	0	1.5 ^a^	0	6	**+**	**+**
I+Cwax	0	4	2	0	1 ^a^	0	6	**−**	**−**
Cwax	0	4	2	0	1 ^a^	0	6	**−**	**−**

**Table 2 jfb-14-00507-t002:** Proportion of the percentage of mature/immature collagen fiber of each group. * values > 1.00 indicate presence of more than 50% of mature fiber; different superscript letters indicate statistical difference.

Groups	7D	15D	30D
C	0.619 ± 0.237 ^a,b^	0.805 ± 0.189 ^a^	0.724 ± 0.556 ^a,b^
CH+P	0.357 ± 0.365 ^a^	0.400 ± 0.254 ^a,b^	1.539 ± 0.707 ^a,^*
CH+P+I	0.486 ± 0.604 ^a,b^	0.394 ± 0.473 ^a,b^	0.559 ± 0.396 ^b^
I+Cwax	0.874 ± 1.450 ^b^	0.321 ± 0.2075 ^b^	0.605 ± 0.393 ^a,b^
Cwax	0.408 ± 0.506 ^a^	0.601 ± 0.315 ^a,b^	0.088 ± 0.405 ^b^

## Data Availability

The data presented in this study are available on request from the corresponding author.

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
