# Peer review of "Biocompatibility, Biomineralization and Induction of Collagen Maturation with the Use of Calcium Hydroxide and Iodoform Intracanal Dressing"

_jfb, 2023, doi:10.3390/jfb14100507_

Round 1

Reviewer 1 Report

Biocompatibility, biomineralization and collagen maturation induced by calcium hydroxide and iodoform intracanal dressing.

The authors nicely conducted an histological studies on diffent intracanal dressings, produced by varyng the percentages of Calcium hydroxide, iodoform and carbowax.

Biocompatibility, biomineralization and collagen maturation were assessed. Overall the work has been well conducted. The main concern is regarding the clinical applications of the data provided by the authors. An intracanal dressing, per definition, is strictly used INTO the canal, while extrusion of this is not recommended. Therefore I cannot find a biological rationale of analysijg the collagen maturation and biomineralization potential.

In addition, biomineralization requires a bioactive component (such as CaSi).
On the contrary, Antimicrobial and Calcium release could be effective properties of the designed materials.

I suggest to add some limitations to better specify these concepts.

-Please explain that biomineralization requires a bioactive components. Previous studies used Calcium silicate based materials, or Bioglass in order to have the nucleation of amorphous calcium phosphates and apatite. Find papers regarding related to endodontics that explain this concept and add this in the discussion.

In particular, the formation of calcium carbonate does not act as a mineralization nucleus. Recent papers assessed the Chemical-Physical Properties and Bioactivity of New Premixed Calcium Silicate-Bioceramic Root Canal Sealers and reported that the formation of calcium carbonate slowered or hindered the nucleation of Apatite.

english is fine. 

Author Response

We are grateful for Reviewer #1 for the chance to improve our paper and appreciate the opportunity to discuss our research. All considerations were responded in the attached file.

Reviewer 2 Report

Authors have conducted this in vivo study to investigate the biocompatibility, inflammatory response, and biomineralization capabilities of calcium hydroxide, iodoform, and different modifications of these two. 

I have mentioned my issues and concerns with this study down below. Overall, I believe that the whole rational behind this study on an outdated material (calcium hydroxide) has not been fully explained nor displayed. I find the surgical method of this in vivo study extremely confusing; authors need to provide more explanation on to why they chose this method instead of a direct implementation of the materials subcutaneously. Furthermore, authors need to include more explanation as to why they chose this site for their examination and not any other site, and why they did not run their experiments on the teeth and decided to place their materials in an ectopic site. Authors need to provide convincing and extensive answers for all of these questions. 

Introduction:

Line 32:

With all due respect, I do not fully agree with this statement. Nowadays, with the tremendous progress that we have observed in the endodontics and endodontic materials/cements, the main objective of endodontic treatments is not just preventing apical periodontitis. Curing or preventing apical periodontitis through endodontic treatments has been done successfully for decades and it is not a new concept. On the other hand the regenerative aspect of the endodontic treatments has been the main focus of research and product development in the past years. I find the first 3 lines of the introduction (Lines 32, 33, and 34), extremely superficial and shallow. I highly recommend authors do a better and more extensive search on the current status of the regenerative endodontics in the literature and provide fulfilling statements in their introduction that actually teaches the readers/reviewers something new, instead of a plain repeat of the already well-known facts. And also only 1 reference for these 3 lines with shallow statements is questionable. I suggest authors restructure the first paragraph of their introduction (which arguably must be the most intriguing part of any introduction) and include a lot more of newly-published studies with accurate and spot-on statements. 

Line 36:

“Effectively root canal ….” Is grammatically incorrect, use “effective” instead.

My main issue with the introduction and the whole objective of this study, is the fact that authors have not raised any questions and have not prepared nay statements extracted from newly-published studies that supports their opinion that calcium hydroxide is still a clinically successful intracanal dressing compared to newer materials such as hydraulic calcium silicate-based cements (hCSCs).

The fact that authors have jumped right into their objectives of this study without first explaining to and convincing the readers that calcium hydroxide has comparable outcomes to hCSCs and other modern materials or not. It is like authors expect readers to agree that calcium hydroxide has a lot of biocompatible values. Which personally, I do not agree with. I believe that hCSCs have outperformed calcium hydroxide in almost all kinds of endodontic treatments, specifically when the main focus of the experiment was the regenerative capabilities of these materials. 

And overall, I believe that for the reasons I mentioned, the length of this introduction is very short and underwhelming. Since the field of endodontic materials and cements has generated numerous hot topics and debates, authors must provide multiple newly-published studies that supports their statements and opinions. They also must include studies that dispute their claims and by doing so authors have to prove to their readers that even though there multiple groups of modern materials such as hCSCs, calcium hydroxide still has enough capabilities to be tested in vivo. Otherwise this study and all of its surgeries would be pointless.

As I mentioned before, this introduction is simply not acceptable for a subject that has countless published in vivo and in vitro studies. A lot of the 18 references used in introduction are outdated and simply not reliable. I highly suggest authors extend their introduction to convince readers of their rationale, and also include more and much newer and more reliable studies as their references in this section.

Methods and Materials:

While I highly respect the detailed reported steps of their material preparation and surgical methods, I still do not understand the necessity or the rationale behind implementing these tubes and not the materials themselves. 

Authors could have had used these materials when they were fully set, so that the surrounding tissues would have had direct contact with these materials.

And if authors were trying to demonstrate an “indirect contact” approach, then it must be further explained why they chose this method.

Overall, I find the surgical method of this in vivo study very confusing and insufficient.

There are a few grammatical and punctuation errors in this study. Authors must use "i.e.," and "e.g.," instead of plain paratheses in their manuscript. 

Author Response

We are grateful for Reviewer #2 for the chance to improve our paper and appreciate the opportunity to discuss our research. All considerations were responded in the attached file.

Reviewer 3 Report

The main question of this research is to verify the difference between iodoform and calcium hydroxide pastes regarding inflammation, biomineraliz tion and collagen fiber maturation..

The topic is interesting and original in the field. The research methodology was applied correctly and carefully. The conclusions consistent with the evidence and arguments presented. All the reference are appropriates.

Authors should include a paragraph explaining the clinical applications of the research, and an other paragraph explaining the subject area compared with other published research.

I belive that with small modify the article could be accepted for publishing

Author Response

We are grateful for Reviewer #3 for the chance to improve our paper and appreciate the opportunity to discuss our research. All considerations were responded in the attached file.

Reviewer 4 Report

The study topic appears to be of clinical interest. However, a number of concerns need to be addressed.

Title: Please rephrase the title. “Biocompatibility, biomineralization, and induction of collagen maturation of calcium hydroxide and iodoform intracanal dressing”. Like so.

Abstract: OK. But no need to use capital letters for polarized light (entire manuscript).

Introduction: OK.

Mat & Methods: In 4th paragraph add extra information on how you prepared the test materials. And also add manufacturers info in parenthesis. In Statistical Analysis Subheading, After performing normality analysis, which tests did you conducted? Add this information.

Results: OK. Ä°n legends of the figures please correct the multiplyer symbol. Before the value not after.

Discussion: Please add study limitations.

Conclusion: OK.

References: Ok.

Sincerely,

Spelling errors, especially those related to the use of spaces and spaces before symbols, should be eliminated.

Author Response

We are grateful for Reviewer #4 for the chance to improve our paper and appreciate the opportunity to discuss our research. All considerations were responded in the attached file.

Round 2

Reviewer 1 Report

The authors answered to all issues. It can be accepted now.

Congratulations

Author Response

We are grateful for all considerations and improvement of our manuscript. 

Reviewer 2 Report

My biggest issue with this paper in my initial review, was the lack of necessity/novelty with this topic. I thought authors wanted to focus on the regenerative part of their experimented materials, yet they main goal was preventing and treating apical periodontitis and regenerative medicine was not their main focus. Authors have provided sufficient answers to all of my questions and concerns. The English grammatical errors have been solved and the authors have promised that if their paper gets accepted they will send their manuscript for English grammar and vocabulary checks in MDPI.

I sincerely appreciate the extremely detailed responses that the authors have provided in their revision. I believe with all of the applied changes, this paper is finally ready to be published.

The English grammatical errors have been solved and the authors have promised that if their paper gets accepted they will send their manuscript for English grammar and vocabulary checks in MDPI.

Author Response

We are grateful for all considerations and your recognition in our extensive response to every question pointed by the reviewer. The english writing in the manuscript was revised. According to the report, in all manuscript there were several minor gramatical errors, such as excessive use of the passive voice, excessive use of the definite article "the", and a few misspelling we didn't notice. The text has become more fluid, making it easier to read.